# Air temperature partitioning of snow accumulation, erosion and melt: a regime shift occurring on Mt. Ortles (Eastern Italian Alps)

Tiziana Lazzarina Zendrini<sup>1</sup>, Luca Carturan<sup>1</sup>, Michael Lehning<sup>2,3</sup>, Mathias Bavay<sup>2</sup>, Federico Cazorzi<sup>4</sup>, Paolo Gabrielli<sup>5</sup>, Nander Wever<sup>2</sup> and Giancarlo Dalla Fontana<sup>1</sup>

<sup>1</sup>Department of Land, Environment, Agriculture and Forestry, University of Padua, Legnaro, Italy

Correspondence to: Tiziana Lazzarina Zendrini (tizianalazzarina.zendrini@phd.unipd.it)

Abstract. Glacier mass balance measurements and models are key tools for understanding the glacier response to climate change and specific processes occurring at the glacier surface. Snow accumulation and wind-driven erosion are among the most difficult processes to measure and model in high-altitude alpine terrain and on glaciers, due to their high variability in space and time, and to the scarcity of in situ observations. Here we use a unique dataset of nivo-meteorological and mass balance observations collected between 2011 and 2015 at 3830 m a.s.l. on Mt. Ortles (Eastern Alps) to investigate snow accumulation and erosion processes. We applied the physics-based snow cover model SNOWPACK, constrained by field data, to reproduce the local mass balance and to explicitly simulate snow erosion by wind. The model reproduces the observed seasonal and annual mass balance variability with good accuracy over the four-year study period. Results indicate that wind erosion is the dominant ablation process at the study site, removing 21% of the snowfall, whereas melt plays a minor role. Erosion is most effective in winter, during or shortly after snowfall events, and its efficiency is controlled by air temperature, with dry snow being much more susceptible to erosion than wet snow. Sensitivity experiments to air temperature perturbations demonstrate that wind erosion provides a negative feedback to the mass balance, because increasing temperature accelerates snow metamorphism and makes the snow surface less erodible. However, a further 1°C warming would promote a transition from an erosion-dominated to a melt-dominated mass balance regime. Our findings emphasize the importance of accounting for wind erosion in projections of glacier mass balance under climate change. They also highlight the relevance of snow erosion for the interpretation of ice core records, because long-term variations in snow erosion may have affected the formation and preservation of the seasonal paleoclimatic signal.

#### 1 Introduction

The local mass balance dynamics of alpine glaciers are characterised by high variability in time and space, resulting from the interaction between atmospheric processes and the local topography. In addition, the glacier mass balance is regulated by multiple feedback mechanisms. Powerful positive feedbacks are the glacier cooling effect (Carturan et al., 2013; Shaw et al.,

<sup>&</sup>lt;sup>2</sup>WSL Institute for Snow and Avalanche Research SLF, Davos, Switzerland

<sup>&</sup>lt;sup>3</sup>CRYOS, School of Architecture, Civil and Environmental Engineering, EPFL, Lausanne, Switzerland

<sup>&</sup>lt;sup>4</sup>Department of Agriculture and Environmental Sciences, University of Udine, Udine, Italy

<sup>10 &</sup>lt;sup>5</sup>Italian Glaciological Committee c/o University of Turin, Turin, Italy

2023, 2025), albedo (Klok and Oerlemans, 2004; Johnson and Rupper, 2020), and elevation (Schäfer et al., 2017). Negative feedback are represented by debris and avalanche accumulation (Benn and Evans, 2014; Capt et al., 2016), topographic shadowing (De Marco et al, 2020), cloud cover (Zhao et al., 2023). These mechanisms add complexity in the glacier mass balance response to atmospheric changes. Despite good knowledge obtained for single processes, accounting for their interactions in glacier mass balance models or in paleoclimatic reconstructions from glacier recorded data (e.g. ice cores) is highly challenging, also because they lead to non-linearities in the climatic response of glacier systems (Ayala et al., 2015; Carrivick et al., 2015; Hock and Huss, 2021).

A main source of uncertainty affecting glacier mass balance models, or reconstructions from proxy data, concerns snow accumulation (Clark et al., 2011; Hock et al., 2017), which typically exhibits complex spatial patterns and high time variability, especially at high-elevation and wind-exposed areas. Wind action is indeed responsible for preferential snow deposition (Lehning et al., 2008), erosion and redistribution, which result in a spatial distribution of snow which is strongly dependent on wind speed, direction and interaction with the local topography (Clark et al., 2011; Mott et al., 2011, 2018).

High-elevation and wind-exposed ice caps and saddles in the accumulation areas of glaciers are often selected for ice core drilling, for paleoclimatic and paleoenvironmental reconstructions. These sites are suitable for ice core drilling for the dominant vertical-minimum horizontal flow of the snow/ice but also because of low air temperature, which preserves climatic and environmental proxy data from melt water percolation, and low snow accumulation rate (due to snow redistribution by wind), which enable recording paleoclimatic and paleoenvironmental data spanning periods of time from seasonal to multimillennial time scales in the Alps (Konrad et al., 2013; Gabrielli et al., 2016; Bohleber et al., 2019). However, dating and reconstructing paleoenvironmental conditions from ice and firn cores extracted at these sites may be biased, because wind can remove a large fraction of the total accumulated snow, potentially erasing precipitation that occurred over entire months or seasons (e.g. Lehning et al., 2008; this work).

Proxy system models (models that describe the processes by which environmental conditions are recorded in a glacial archive) are often used to improve dating and reconstruction of past climatic conditions obtained from ice/firn cores (e.g. Evans et al., 2013). For instance, these models calculate how stable water isotopes are recorded in ice core archives, generating a pseudo proxy that is compared to the actual water isotopes, to constrain its paleoclimatic interpretation (typically air temperature). These models implicitly assume negligible year-to-year variation of snow redistribution, melt, and meltwater percolation, meaning that they do not explicitly calculate the glacier mass balance and do not take into account its variability over time.

Low latitude drilling sites are increasingly affected by atmospheric warming and loss of seasonal or even annual snow accumulation due to enhanced melt. Considering these conditions, Carturan et al. (2025) proposed a proxy system model that explicitly accounts for the glacier mass balance variability over time. When compared to traditional annual layer counting of stable isotope and pollen seasonal oscillations for producing an ice core chronology, the authors found that the model significantly improved the interpretation of the firn stratigraphy during the 1996-2011 warm period at the high-altitude drilling site of Mt. Ortles, in the Eastern Alps. Among possible further developments of their model, Carturan et al. (2025) mentioned a simple parameterization of snow erosion and its dependence on air temperature.

In this context, a critical simplification affecting most glacier mass balance models regards the calculation of snow accumulation and wind redistribution, and in particular their long-term response to variations in air temperature/wind intensity. Snow accumulation is generally handled statistically, using precipitation data recorded by automatic weather stations. Precipitations are extrapolated on glaciers using multiplication parameters that account for the vertical gradients of precipitation and its redistribution. These parameters are tuned using available observations of mass balance and/or remote sensing-derived snow cover (e.g. Schuler et al., 2005; Huss et al., 2008; Carturan et al., 2012; Cremona et al., 2025) and are assumed to be constant over time. This approach is normally used due to the scarcity of direct observations of snow redistribution. It implicitly assumes that the relationship between low-elevation precipitation and snow accumulation on glaciers is fixed over time. However, as suggested by Haeberli and Alean (1985) and reported by Li and Pomeroy (1997a) and He and Ohara (2017), there is strong evidence that air temperature regulates the susceptibility of the snowpack to wind erosion. The wind speed threshold for initial motion is controlled by snow metamorphism that, along with other factors such as snow wetness, kinetic friction, and elasticity, is strongly influenced by air temperature (Li & Pomeroy, 1997b).

As a result, the role of air temperature on snow susceptibility to wind erosion can be considered an important feedback that regulates snow accumulation from seasonal to centennial and millennial time scales, especially on high-elevation and wind-exposed accumulation areas of high-altitude glaciers. Because cold and dry snow at high elevations is more easily eroded by wind than temperate snow (often containing refrozen melt layers), it is possible that the annual snow accumulation rate and/or the seasonal magnitude of snow erosion are changing as a response to atmospheric warming (Haeberli and Alean, 1985). In this case, paleoclimatic reconstruction from ice/firn cores would significantly benefit from improved modelling of the relationship between snow erosion and air temperature/wind conditions.

In this paper, we investigate the snow processes observed at an automatic weather station operated between 2011 and 2015 at 3830 m a.s.l., next to the Mt. Ortles drilling site at 3859 m a.s.l., where detailed meteorological and snow observations are available (Carturan et al., 2023). Using the physics-based process-oriented SNOWPACK model (Lehning et al., 2002a, 2002b), which explicitly accounts for snow erosion by wind, we characterise the snow erosion process, its variability over time, and its dependency on meteorological variables, particularly air temperature. Finally, we analyse the climatic sensitivity of net snow accumulation and erosion for six different scenarios of atmospheric warming and cooling.

#### 2 Study area

The study area is located on Mt. Ortles (3905 m a.s.l.;  $46.508^{\circ}$ N,  $10.541^{\circ}$ E), the highest peak in the Eastern Alps, within the Ortles-Cevedale Mountain Group. The Alto dell'Ortles glacier lies on the north-western slope of the mount, extending from 3018 to 3905 m a.s.l. and covering an area of 1.19 km² (2017). Ice thickness reaches up to  $\sim$ 75 m (Gabrielli et al., 2012), with the basal ice layers preserving a paleoclimatic record of the past  $\sim$ 7 kyr (Gabrielli et al., 2016, 2025). The glacier exhibits a polythermal structure, with temperate firn overlying colder ice below the firn-ice transition at  $\sim$  30 m depth (Gabrielli et al., 2012).

Since 2008, this glacier has been the focus of the "Ortles Project" (ortles.org), an international research program aimed at recovering deep ice cores for paleoclimate reconstruction and collecting data for multiple cryospheric components.

Mt. Ortles is located near the inner dry zone of the Alps and is subject to a continental climate. Long-term observations from Solda (valley floor) indicate a mean annual precipitation of 800–950 mm between 1981 and 2010 (Adler, 2015). On the top of Mt. Ortles, annual precipitation is estimated between 1300 and 1400 mm of water equivalent, based on mass balance observation conducted from 2009 to 2016 (Carturan et al., 2023). This precipitation estimate may vary significantly across space due to the influence of the wind on snow accumulation and redistribution.

At 3850 m a.s.l. the mean annual air temperature is approximately -9°C. Snow accumulation occurs mainly from October to May. The melt season typically spans from June to September, though summer snowfalls are frequent on the glacier summit. The intensity and duration of ablation exhibit substantial differences from year to year and are strongly influenced by heatwaves. Rainfall is rare at these elevations, but isolated rain events have been observed on the summit over the past 15 years (Carturan et al., 2023).

In the period of direct observations, between September 2011 and June 2015, the mean monthly air temperature ranged between -17.24 and -1.63°C, peaking in August (Fig. 2a). The precipitation showed a distinct maximum in summer and a minimum in winter, which is typical of the local climate (Fig. 2b). The wind speed was highest in winter and lowest in spring and in September (Fig. 2c). The dominant wind direction was from south-west (Fig. 2d).

115

100

105

Figure 1: Location of the ice core drilling site and the automatic weather station (AWS) on Mt. Ortles. The background hill-shaded DEM (2017 lidar survey) is from https://mapview.civis.bz.it (last access: 15 September 2025) (Agenzia per la Protezione civile, Autonomous Province of Bolzano).

Figure 2: Monthly means of (a) air temperature, (b) precipitation, (c) wind speed, and (d) wind direction calculated in the period between September 2011 and June 2015. Air temperature and wind data come from the automatic weather station at 3830 m a.s.l., whereas precipitation data come from the Solda village, right below Mt. Ortles, at 1907 m a.s.l.

## 3 Methods

## 3.1 Meteorological observations

An automatic weather station (AWS) operated in the upper accumulation zone of the Alto dell'Ortles Glacier from September 2011 to June 2015. The AWS was installed at 3830 m a.s.l., approximately 200 m downslope of the drilling site, and on a site with western exposure. A Campbell Scientific CR-1000 data logger stored 15-minute values coming from sensors of air temperature and relative humidity (Vaisala HMP155A), wind speed and direction (R. M. Young 05103), incoming and outgoing shortwave and longwave radiation (Delta Ohm LP Pyra 05 and LP PIRG 01), and snow depth (Campbell Scientific SR50A). Details on sensors, radiation shields, and measured data can be found in Carturan et al. (2023). The AWS was powered by solar panels.

The AWS was mounted on an aluminium tower, composed of two-meter segments and anchored two meters deep into the snow, supported by wooden platforms at the bottom. Once installed, the tower rose 4 meters above the snow surface. Sensors and photovoltaic panels were placed on the top of the structure, while the data loggers and batteries were housed in a fiberglass box just above the snow surface. Because of the snow accumulation, the height of the sensors relative to the glacier surface varied seasonally. To ensure continuous operation and avoid sensor burial by net accumulation of snow, the tower was extended each year by adding a two-meter segment each time.

Precipitation could not be measured with reasonable accuracy at the Ortles AWS, because it would have required a heated rain gauge with too high-power demand. In addition, due to the high elevation and wind exposure, any data collected would have been highly underestimated and difficult to correct, due to the well-known issues (high wind speed, high frequency of snowfall, rain gauge perturbations of the wind field) affecting precipitation measurements in similar environments (Rasmussen et al., 2012; Kochendorfer et al, 2017). Consequently, a rain gauge was not installed at the AWS, using and scaling instead precipitation data measured at the nearest weather station in Solda (1907 m a.s.l.), operated by the local meteorological agency (Ufficio Idrografico, Provincia Autonoma di Bolzano, meteo.provincia.bz.it).

The raw meteorological data from Mt. Ortles and Solda were quality-checked and validated against observations from neighbouring weather stations (details are reported in Carturan et al. 2023), specifically Madriccio (2825 m) and Cima Beltovo (3328 m) for air temperature and wind speed, and Fontana Bianca (1900 m) for precipitation. Precipitation data were also corrected following the procedure described in Carturan et al. (2012,) to account for the rain gauge underestimation. Since the SNOWPACK model requires a continuous meteorological input in the investigated period, missing or unreliable data were reconstructed using backup sensors at the AWS when available, and data from Madriccio and Cima Beltovo weather stations in case backup sensors were unavailable or unreliable. Standard gap-filling methodologies implemented within the MeteoIO/SNOWPACK toolchain were applied (Section 3.3, Bavay and Egger, 2014).

#### 3.2 Mass balance observations

Seasonal and annual glacier mass balance measurements were carried out between June 2009 and September 2014 at the ice core drilling site and at the AWS location. Winter balance surveys were performed in June or early July before the start of the ablation season. Summer and annual balance measurements took place in late August or early September, at the end of the melt season. Field observations included snow depth probing around both sites, combined with snow and firn density measurements in snow pits dug down to the previous summer surface. Snow stratigraphic profiles were carried out along shaded pit walls, including observations of snow and firn temperature, hardness, grain type and size, and the presence of ice lenses and dust layers (Gabrielli et al., 2010). These stratigraphic observations were helpful in identifying the summer surface, both within snow pits and while probing snow depth. Density measurements were used to convert snow depth into water equivalent.

#### 3.3 Snow modelling

We used the snow cover model SNOWPACK (Lehning et al., 2002a, 2002b) to calculate the energy and mass balance at the 165 Ortles AWS site. Originally developed for avalanche warning, SNOWPACK is widely used to study the alpine cryosphere, including snow and glacier mass balance and mountain permafrost (Meirold-Mautner and Lehning, 2004; Obleitner and Lehning, 2004; Rasmus et al., 2004; Luetschg et al., 2008; Bavay et al., 2009, 2014). SNOWPACK is a physics-based, onedimensional, multi-layer numerical model designed to simulate the evolution of the snow cover using meteorological data 170 from automatic weather stations. The model simulates the stratigraphy and the microstructure of the snow by modelling energy and mass fluxes. It employs a Lagrangian finite-element method to solve the heat transfer equations. A key feature of SNOWPACK is its detailed representation of snow metamorphism, which is closely linked to mechanical properties such as the thermal conductivity and viscosity. The model also accounts for the interactions between the snow surface and the atmosphere, including radiative transfer processes such as the penetration of shortwave radiation. Its computational framework 175 is based on a set of key state variables that serve as the foundation for simulating the snow microstructure and its metamorphic changes. These variables are used to derive essential bulk properties needed to solve the model core conservation equations. A detailed description of the mathematical formulations and governing equations implemented in SNOWPACK can be found in Bartelt and Lehning (2002), and in Lehning et al. (2002a, 2002b).

SNOWPACK explicitly accounts for wind-driven snow erosion through its snowdrift module (Lehning et al., 2000). The scheme first determines a threshold friction velocity (m s<sup>-1</sup>) at the snow surface following Schmidt (1980), based on its erodibility calculated as a function of snow grain shape, size and bond strength (Lehning et. al, 2000). When the actual friction velocity, derived from the logarithmic wind profile including stability corrections, exceeds the threshold friction velocity, snow erosion initiates. The horizontal drifting snow mass flux is then calculated following Lehning and Fierz (2008). To translate the horizontal mass flux into a vertical surface erosion mass flux, a fetch length of 70 m is assumed (Lehning and Fierz 2008, Wever et al. 2023). At each time step, the erodibility of the surface layer is updated, allowing SNOWPACK to consistently represent erosion also during snowfall. To run the model the following meteorological variables were used as inputs: air temperature, relative humidity, wind speed, incoming and outgoing shortwave radiation, incoming and outgoing longwave radiation and precipitation. The model framework relies on the MeteolO library (Bavay and Egger, 2014) for meteorological data pre-processing. This module acts as a middleware layer between raw meteorological data and the model, pre-processing data input from various sources, applying quality control, filtering, resampling and interpolation to provide consistent and reliable forcing data.

A point simulation of the snowpack was conducted at the AWS site using a 15-minute time step interval. The simulation began on 7 September 2011 and extended until 14 of June 2015. SNOWPACK was initialized using snow pit observations collected at the AWS site a few days before the start of the simulation. These observations extended from the glacier surface down to 400 cm and included snow and firn stratigraphy (grain size and shape), density, and the temperature profile. The snow accumulation was simulated using the Solda precipitation, multiplied by a correction factor to account for the increase of

precipitation with elevation (vertical precipitation lapse rate). The wind speed was also adjusted by means of a multiplicative factor, to account for the fact that on top of Mt. Ortles the wind is not expected to follow strictly a vertical logarithmic profile (Mott et al., 2010; Stiperski and Rotach, 2016). The multiplicative factors for precipitation and wind speed were adjusted iteratively to minimize the Root Mean Square Error (RMSE) between measured and modelled mass balance at the simulation site. We constrained the precipitation lapse rate using local values published by Carturan et al. (2019).

#### 3.4 Analysis of snow erosion

The snow erosion process was analysed in terms of snow water equivalent (SWE), rather than in terms of snow height, allowing changes due to wind redistribution to be isolated from those caused by other processes, like compaction. We excluded from analyses the periods with interpolated wind speed data (7% of the total, Table B1), to avoid introducing possible external noise to the data.

The analyses regarded hourly snow erosion events, identified using the SNOWPACK output variable MS\_Wind (mm h<sup>-1</sup>), which describes the mass loss during erosion. An hourly snow erosion event was identified when MS\_wind exceeded 0.1 mm h<sup>-1</sup>. Each hourly snow erosion event was characterized using descriptors that quantify the meteorological conditions during erosion, and the average conditions during the life of each snow layer that was removed by wind erosion (Table 1). The beginning of each layer's life was set when snow accumulation exceeded 0.1 mm h<sup>-1</sup>.

The snow erosion process was analysed using correlation and frequency distribution analyses. Two correlation matrices were calculated: (i) a matrix including all hourly erosion events (n = 1886), and (ii) a matrix that includes the WS\_Th and T\_Th variables, which could be calculated only when erosion starts (i.e. they could not be calculated over consecutive hourly erosion events; n = 588).

Because the mechanical response of snow to wind differs under wet and dry conditions, erosion events were classified accordingly. Wet snow was defined as snow exposed to positive air temperatures since layer formation, representing a cohesive wet regime. Dry snow, in contrast, was defined as snow that never experienced positive air temperatures. Based on this criterion, 1731 dry snow erosion events and 155 wet snow erosion events were identified.

Table 1: Calculated descriptors used to characterize the hourly snow erosion events.

| Variable name                        | Unit                   | Explanation                                                                           |
|--------------------------------------|------------------------|---------------------------------------------------------------------------------------|
| WE_Eroded (Water equivalent eroded)  | mm                     | Total SWE loss during the event                                                       |
| WS_Erosion (Wind speed erosion)      | $m \ s^{-1}$           | Wind speed during the event                                                           |
| T_Erosion (Air Temperature Erosion)  | °C                     | Air temperature during the event                                                      |
| WS_Th (Wind speed threshold)         | ${\rm m}~{\rm s}^{-1}$ | Mean wind speed between the hour preceding erosion and the first hour of erosion      |
| T_Th (Air temperature threshold)     | °C                     | Mean air temperature between the hour preceding erosion and the first hour of erosion |
| Age (Age)                            | h                      | Time since the formation of the snow layer                                            |
| WS_Life (Wind speed life)            | $m \ s^{-1}$           | Mean wind speed since layer formation                                                 |
| T_Life (Air temperature life)        | °C                     | Mean air temperature since layer formation                                            |
| PDG_Life (Positive degree days life) | °C                     | Cumulative positive degree hours since layer formation                                |
| MELT_Life (Melt life)                | mm                     | Cumulative SWE lost by melt since layer formation                                     |

## 3.5 Sensitivity analysis of snow erosion to changes in air temperature

To assess the sensitivity of wind-driven snow erosion to changes in air temperature, we performed additional SNOWPACK simulations applying a fixed offset to the input temperature data. Six scenarios were simulated, shifting the observed air temperature data by -3, -2, -1, +1, +2, and +3 °C, relative to the baseline (i.e. the observation period from 2011 to 2015). Each experiment covered the same period as the reference run (7 September 2011 to 14 June 2015). This approach, even if simplified, aims to investigate the response to both colder conditions (representative of past centuries) and plausible near-future warming at high elevation in the Alps (IPCC, 2022). The precipitation data were left unchanged, as no significant long-term trends have been detected in the study area (De Blasi et al., 2018), nor do future projections indicate clear trends for this geographic area (IPCC, 2022). For each scenario, we analysed the model output in terms of snow water equivalent, the change

in the relative contribution of erosion, melt and surviving snow to the total mass balance, and the change in the monthly regime of snowfall erosion by wind.

#### 4 Results

235

#### 4.1 Snowpack mass balance modelling

The RMSE between measured and modelled mass balance at the simulation site was optimized using a multiplicative factor for precipitation measured at Solda of 1.55 and a multiplicative factor for wind speed of 0.70. Considering all mass balance measurement sub-periods (Table 2), the RMSE between measured and modelled SWE above the previous year's summer surface is 0.115 m w.e.

According to SNOWPACK calculations, 4.401 m w.e. accumulated at the study site between 17 Sep 2011 and 23 Sep 2014 (Table 2). The annual accumulation rate was lower in the 2011/12 balance year and higher in the 2012/13 and 2013/14 balance years, with similar rates in the latter two years. The largest part of snow accumulation occurred during the cold season (from October to March), while the warm season (from April to September) still contributed, with summer gains ranging from 0.04 m w.e. (2012) to 0.49 m w.e. (2014), whereas 2013 experienced a net summer ablation of 0.24 m w.e.

Table 2: Observed and modelled snow water equivalent (SWE) above the previous year's summer surface and cumulated balance at the Mt. Ortles AWS site.

|                  | Observ                                                | ations                        | Simulation                                      |                               |  |  |
|------------------|-------------------------------------------------------|-------------------------------|-------------------------------------------------|-------------------------------|--|--|
| Measurement date | SWE above the previous year's summer surface (m w.e.) | Cumulated balance<br>(m w.e.) | SWE above the previous year's summer surface (m | Cumulated balance<br>(m w.e.) |  |  |
| 17/09/2011       | 0                                                     | 0                             | 0                                               | 0                             |  |  |
| 18/06/2012       | 0.748                                                 | 0.748                         | 0.763                                           | 0.763                         |  |  |
| 25/08/2012       | 0.792                                                 | 0.792                         | 0.919                                           | 0.919                         |  |  |
| 01/07/2013       | 1.474                                                 | 2.266                         | 1.348                                           | 2.267                         |  |  |
| 08/08/2013       | 1.231                                                 | 2.023                         | 1.433                                           | 2.352                         |  |  |
| 03/07/2014       | 1.496                                                 | 3.519                         | 1.495                                           | 3.803                         |  |  |
| 23/09/2014       | 1.988                                                 | 4.011                         | 2.049                                           | 4.401                         |  |  |

#### 4.2 Analysis of snow erosion

## 4.2.1 Correlation analysis

In addition to the dependency of WE\_Eroded on WS\_Erosion, both correlation matrices indicate that WE\_Eroded is positively correlated also with WS\_Life and WS\_Th (Figs. 3 and 4). Interestingly, WS\_Life is strongly anticorrelated with age, suggesting that wind speed is highest during precipitation events, or shortly afterwards. Age is positively correlated with WS\_Th and WS\_Erosion, meaning that older snow layers require higher wind speed to start being eroded. Age is also positively correlated with PDG\_Life and MELT\_Life, suggesting that the longer a layer persists, the higher the probability that it experiences positive temperature. In contrast, Age is negatively correlated with T\_Erosion and T\_life, indicating that layer persistence for long periods, before being eroded, is more likely during winter.

The erosion rate (WE\_Eroded) does not look much influenced by temperature-related variables (T\_Th, T\_Erosion, T\_Life, PDG\_Life, MELT\_Life). This means that air temperature does not directly influence the snow erosion rate once the erosion has started. However, it strongly influences initiation and frequency of erosion, as reported in section 4.2.2 (Figs. 7 and 8).

Figure 3: Spearman correlation matrix among erosion-event descriptors for all erosion hours (n = 1886). Colour encodes sign and magnitude of correlation coefficients (blue = negative, red = positive). Asterisks indicate significance (\* p 

Figure 4: Spearman correlation matrix among erosion-event descriptors restricted to hours when erosion starts (n = 588). Colour encodes sign and magnitude of correlation coefficients (blue = negative, red = positive). Asterisks indicate significance (\* p < 0.05, \*\* p < 0.01, \*\*\* p < 0.001).

In Fig. 5 clear differences emerge between the hourly erosion rates of wet and dry snow. The hourly events are also split into 'during snowfall' and 'after snowfall' and fitted with four cubic functions passing through the origin. During snowfall, the dry snow fit lies significantly above the wet snow fit. For example, at a wind speed of 15 m s<sup>-1</sup> the erosion rate is 0.95 mm w.e. h<sup>-1</sup> for dry snow (51% higher) when compared to 0.63 mm w.e. h<sup>-1</sup> for wet snow. After snowfall, however, the dry and wet

285

snow erosion rates are very similar. Differences between erosion rates during and after snowfall are much larger for the dry snow, compared to wet snow.

Approximately 91% of the erosion events removes less than 1.5 mm  $h^{-1}$ , whereas the maximum rates exceeded 3.5 mm w.e.  $h^{-1}$ , occurring for wind speeds greater than 18 m  $s^{-1}$ .

Figure 5: Hourly erosion rate versus wind speed with least-squares cubic fits through the origin. Erosion events and fittings are separated into dry/wet snow and during/after snowfall.

## 4.2.2 Frequency distribution analysis

The cumulative relative frequency distributions of snow erosion events and cumulative relative SWE eroded were analysed as a function of layer age to assess when the largest amount of snow erosion takes place. The analysis shows that erosion occurs predominantly soon after deposition (Fig. 6). Considering dry and wet snow erosion together, 50% of the total SWE is eroded within the first 6 h, 66% within 24 h, 76% within 48 h and 90% within 240 h. Dry snow erosion is even faster, with 70% of SWE eroded within 24 h, and 95% within 240 h. In contrast, wet snow erosion is slower: only 22% of SWE is eroded within the first 24 h, and 50% within 240 h. The cumulative relative frequency distributions of snow erosion events and cumulative

relative SWE eroded as a function of layer age are very similar, except for wet erosion events that are characterised by a higher relative cumulative frequency of (small) erosion events up to 950 hours of snow layer age.

In the analysed period, dry snow erosion totals 1091.7 mm w.e., corresponding to 1731 hourly erosion events, whereas wet snow erosion totals 113.1 mm w.e., corresponding to 155 hourly events. Overall, dry snow erosion accounts for 91% of the total modelled erosion.

Figure 6: Cumulative relative frequency distributions and cumulative relative SWE eroded as a function of layer age. Panels show (a) all events, (b) dry snow events, and (c) wet snow events.

Fig. 7a shows that modelled erosion occurs more frequently with dry snow at the surface, compared to wet snow. The frequency of dry snow erosion peaks at a wind speed of 11 m s<sup>-1</sup>, whereas wet snow erosion peaks at 16 m s<sup>-1</sup>. The SWE eroded peaks at larger wind speed, 13 m s<sup>-1</sup> for dry snow and between 16 and 21 m s<sup>-1</sup> for wet snow (Fig. 7b). Fig. 8a compares the hours with erosion to the total number of hours at a given wind speed, highlighting that dry snow erosion is negligible up to 8-9 m s<sup>-1</sup> and wet snow erosion becomes relevant only above 13-14 m s<sup>-1</sup>. Fig. 8b shows that for dry snow the maximum frequency of wind erosion is between air temperatures of -12 and -24°C, decreasing towards 0°C. Wet snow erosion peaks at around 0 and +1°C. In both cases, the frequency of snow erosion drops to zero above +1°C.

Figure 7: (a) Frequency of wind speed during modelled erosion for dry snow and wet snow; (b) total SWE eroded per wind speed class for dry and wet snow.

315

Figure 8: (a) Relative frequency of occurrence of snow erosion for dry and wet snow at various wind speeds (hours with erosion divided by all hours at that wind speed; wind speed  $\geq 25$  m s<sup>-1</sup> is not shown because the sample size is smaller than 10 hours); (b) relative frequency of occurrence of snow erosion for dry and wet snow at various air temperatures (hours with erosion divided by all hours at that air temperature; air temperature bins with sample size  $\leq 10$  hours are not shown).

Coherently with Figs. 7 and 8, the frequency distribution of the threshold wind speed for erosion peaks at 9 m s<sup>-1</sup> for dry snow and between 10 and 15 m s<sup>-1</sup> for wet snow (Fig. 9). For the latter, an unambiguous peak could not be calculated, due to the small sample size of the wet snow erosion subset. The mean threshold wind speed is 11.64 m s<sup>-1</sup> for dry snow and 13.47 m s<sup>-1</sup> for wet snow.

Figure 9: Frequency distributions of the threshold wind speed for erosion for dry and wet snow events.

## 4.3 Sensitivity analysis of snow accumulation and erosion to changes in air temperature

The sensitivity of SNOWPACK simulations to the simulated temperature changes is remarkably higher for warming scenarios, compared to cooling scenarios. (Fig. 10). The modelled mass balance is little affected by warming up to 1°C above baseline conditions, which causes a decrease of 11.5% in the final SWE. However, a larger warming has a much stronger impact, causing a 38.5% smaller final SWE for 2°C warming, and a 71.8% smaller final SWE for 3°C warming. In contrast, cooling scenarios produce a minor increase of 0.4% in the final SWE for a 1°C temperature decrease, whereas the final SWE decreases under larger cooling (-2.2% for 2°C cooling and -3.8% for 3°C cooling).

Under current (baseline) conditions, the mass balance is dominated by snow accumulation and wind erosion, which removed 20.9% of the accumulated snow during the observed period. Melt is comparatively less important and removed only 2.9% of total snow accumulation (Table 3, Fig. 11). Up to 1°C warming, wind erosion remains dominant in removing accumulated snow (18.7%) over melt (12.5%). Larger warming implies strong melt increase and a significant decrease in the relevance of snow erosion (16.5% and 14.9% of the total accumulation for 2°C and 3°C warming, respectively). On the other hand, melt decreases almost to zero with 1°C cooling, whereas the share of total snow accumulation removed by wind erosion increases about 2% for each degree of cooling, reaching 27% for the -3°C scenario.

The seasonal regime of snow erosion is currently characterised by a prominent maximum in winter months, when 48% of monthly snowfall is eroded on average, and a minimum in spring, when 11% of monthly snowfall is eroded on average. In

summer and fall, the wind erosion averages 16% and 17% of the monthly snowfall (Fig. 12, Table A1). The highest sensitivity of wind snow erosion to warming temperature is found in spring and fall, with a maximum decrease of about 61% in March for the +3°C scenario. High sensitivity was calculated in mid-summer, whereas lower sensitivity is visible in the coldest months. On the other hand, colder temperatures affect mainly the months of May, August, September and November. In particular, for August a nearly doubling of wind erosion was calculated for the -3°C scenario. The sensitivity is smaller during winter months and in April and June (Fig. 12).

Figure 10: Modelled SWE at the AWS site for the baseline and temperature change scenarios (-3°C to +3°C). Vertical bars indicates melt and erosion hours.

Table 3. Total precipitation, snowfall, erosion, melt, and net accumulation modelled for baseline and temperature change scenarios (-3 °C to +3 °C).

| Variable                 | Baseline | −1 °C   | <b>-2</b> °C | -3 °C   | +1 °C   | +2 °C   | +3 °C   |
|--------------------------|----------|---------|--------------|---------|---------|---------|---------|
| Total precipitation (mm) | 6141.98  | 6141.98 | 6141.98      | 6141.98 | 6141.98 | 6141.98 | 6141.98 |
| Total snowfall (mm)      | 5977.77  | 6005.80 | 6012.94      | 6022.28 | 5863.75 | 5496.89 | 5135.33 |
| Total snowfall (%)       | 97.33    | 97.78   | 97.90        | 98.05   | 95.47   | 89.50   | 83.61   |
| Total erosion<br>(mm)    | 1248.39  | 1396.29 | 1535.38      | 1624.10 | 1099.12 | 909.10  | 765.68  |
| Total erosion (%)        | 20.88    | 23.25   | 25.53        | 26.97   | 18.74   | 16.54   | 14.91   |
| Total melt (mm)          | 173.73   | 37.66   | 20.20        | 14.08   | 733.91  | 1784.60 | 3084.85 |
| Total melt (%)           | 2.90     | 0.63    | 0.34         | 0.23    | 12.52   | 32.47   | 60.07   |
| Net accumulation (mm)    | 4555.66  | 4571.85 | 4457.36      | 4384.10 | 4030.72 | 2803.19 | 1284.80 |

Figure 11: Modelled partitioning of total precipitation into net accumulation (survived snow), wind-driven erosion, and melt for baseline and temperature change scenarios (-3 °C to +3 °C). Percentages (%) refer to the total accumulated snowfall.

Figure 12: Modelled monthly snow erosion (percentage of monthly snow accumulation) for the baseline and temperature change scenarios (-3 °C to +3 °C).

#### 355 5 Discussion

360

## 5.1 Inaccuracies and limitations

In this work, we used the model SNOWPACK to investigate the snow erosion on top of the Alto dell'Ortles glacier, based on mass balance and nivo-meteorological observations collected between 2011 and 2015. It was not possible to analyse the snow erosion directly from the data collected by the snow-depth sensors, because several other processes are involved in snow depth variations, such as sublimation, melt, compaction, and because the density of the surface layers is unknown. This made impossible to quantify erosion mass fluxes just from automatically measured snow depth and required the application of a physics-based model, constrained by field measurements, to disentangle snow erosion from other processes.

365

The SNOWPACK model reproduces well the measured mass balance variability across four years at the study location, provided wind speed is adjusted to account for deviations from smooth, logarithmic profiles over complex topography (Panofsky and Ming, 1983), and the precipitation measured at the valley floor is corrected for rain gauge errors (Carturan et al., 2012) and for the vertical lapse rate. In absence of field observations regarding the time variability of the vertical wind profile and precipitation lapse rate, we used constant correction factors and adjusted wind and precipitation so that the model matched the measured mass balance. Assuming time invariance of these two factors was a necessary simplification that ensures a good model performance but likely affects single events or periods of simulation.

For example, there is very good correspondence between modelled and measured seasonal balances (Table 2), except for summer 2013, when the model overestimates the mass balance by 0.33 m w.e. In this case, SNOWPACK calculated too much snowfall on Mt. Ortles, which increased surface albedo and inhibited melt. This was due to the seasonal variability of the precipitation lapse rate and type of precipitation. In summer, the study site is mainly affected by local convective precipitation (thunderstorms) that are scattered by nature, making precipitation variability more randomly distributed and less correlated with elevation when compared to frontal precipitation, which is more sensitive to orographic uplift (Adler et al., 2015). Using a constant precipitation correction factor led to significant overestimation of summer snowfall and mass balance in 2013, however possible errors in 2013 mass balance measurements in the field cannot be ruled out as well.

Low accuracy affecting mass balance measurements on Mt. Ortles can be mostly related to the difficult detection of the previous year's summer surface during snow depth soundings. The related deviation from the true value can be potentially large (up to 0.5 m w.e.) but was significantly reduced by regularly digging pits into snow and firn, which enabled detection of the previous year's summer surface with good confidence based on observations of grain shape and size, dust layers and melt/refreeze crusts. This check excluded large uncertainties in the evaluation of the annual snow depth. However, it could not completely remove inaccuracies linked to the spatial variability of snowpack thickness and density. Overall, we estimate a random uncertainty of 100 mm w.e. per mass balance measurement, which lies within the range reported in literature for measurements under similar conditions (Zemp et al., 2013).

Inaccuracies in nivo-meteorological variables measured by the AWS may also have contributed to the total uncertainty of the model's output. Considering these sources of possible uncertainty (Sect. 3.1) and the extreme environmental conditions at the study site, the RMSE between measured and modelled seasonal and cumulated balance (0.115 mm w.e.) looks acceptable. However, we preferred to remove from the presented model results periods with data gap filling to minimise external and spurious sources of variability in the modelled SWE series and snow erosion events analysed in this work, while still preserving a sample size that was sufficient for attaining statistical significance.

Another limitation of our approach concerns the simplification adopted for the sensitivity analysis of snow erosion to changes in air temperature, which applies constant air temperature offsets to the meteorological input used in SNOWPACK. We acknowledge that climate change involves more complex patterns, such as different seasonal trends, changes in circulation patterns and precipitation (e.g. Dumont et al., 2025). A further limitation is the relatively short observation period. This may affect, for example, the representativeness of threshold wind speed values for wet snow erosion (due to the smaller sample size

compared to dry snow), or the robustness of monthly erosion regimes (which fluctuate due to the existence of single months with unusual meteorological behaviour). Nevertheless, in view of how unique is this high-elevation four-year dataset, we consider it suitable to characterize the key processes affecting snow erosion and its temperature sensitivity at the study site.

## 400 5.2 Snow erosion and its sensitivity to air temperature

At the Mt. Ortles AWS site and during the investigated period, snow erosion played a relevant role in regulating the surface mass balance, removing 21% of the total accumulated snowfall. At present, snow erosion (likely associated with sublimation of drifted snow, Dery et al., 1998) represents the dominant ablation process at this site, whereas melt only removed 2.9% of the accumulated snow during the modelled period. The western exposure enhances snow erosion, as it frequently results in the AWS site lying on the windward side of the mountain (Figs. 1, 2d and 13a). Several other studies documented the importance of wind exposure of mountain slopes compared to dominant regional wind fields, as a direct control on local snow accumulation and location of glacier accumulation areas (e.g. Humlum, 1987).

Figure 13: Frequency of wind direction for (a) all hours of observations, (b) hours with precipitation, (c) hours of erosion during precipitation, and (d) hours of erosion after precipitation.

Snow erosion is also promoted by high elevation, which results in air temperature below the freezing level most of the time (Fig. 2a). Air temperature has a dominant effect on the susceptibility to snow erosion at the glacier surface, because it is directly related to snow metamorphism and, in particular, to the formation of bonds between snow crystals and of melt-and-refreeze crusts. Dry snow is consequently more frequently eroded when compared to wet snow, regardless of the wind speed causing drifting (Fig. 7). Above 0°C snow erosion becomes negligible (Fig. 8b).

The threshold wind speed for the initiation of erosion is highly dependent on the snow thermal regime. A wind speed of about 8-9 m s<sup>-1</sup> is required to achieve a frequency of dry snow erosion significantly above zero, while this nearly doubles for wet snow (Fig. 8a). Even considering the average threshold wind speed for snow erosion (11.64 m s<sup>-1</sup> for dry snow and 13.47 m s<sup>-1</sup> for wet snow, Fig. 9) our calculated values lie towards the upper end of the range reported in literature, which is between 4-11 m s<sup>-1</sup> for dry snow and 7-14 m s<sup>-1</sup> for wet snow (Li and Pomeroy, 1997). The high threshold wind speed on Mt. Ortles may depend on the high wind speed during precipitation (6.7 m s<sup>-1</sup> on average), which increases the density and cohesion of fresh snow during deposition (Liston et al., 2017).

Precipitation events represent a source of ice particles that can create initial saltation and consequently a lower threshold for wind transport (Schmidt, 1980). This process is likely responsible of the high amount of snow erosion calculated during

precipitation at the Mt. Ortles AWS. Indeed, more than half of the snow erosion occurs during precipitation (53%), or shortly afterwards (Figs. 5 and 6). This is somehow unexpected because during precipitation the atmospheric wind flow in this region is mostly from South or South-East (Fig. 13b), creating lee-side conditions at the AWS site. In fact, most wind erosion was expected to occur during northerly windstorms that frequently follow atmospheric disturbances in this region, especially during winter. In contrast, what occurs is a prevailing wind erosion by south-easterly or south-westerly winds, during or shortly after weather disturbances (Fig. 13c and d) and well before the wind rotation to north (generally associated with the end of the precipitation events).

Snow erosion on Mt. Ortles is mostly a winter phenomenon, due to the combination of high wind speed and low temperature (Fig. 2a and b). On average, 48% of the accumulated snow was removed by erosion in winter, whereas this amount ranges between 10% (in March and May) and 21% (in October) during the other parts of the year. The monthly regime of precipitation, with a maximum in the warm season (Fig. 2c), was more favourable to snow preservation than it could have been if precipitation were equally distributed over the year, or with a winter maximum. Under these conditions, the AWS site would have experienced a significantly lower annual snow accumulation.

The sensitivity of snow erosion to temperature fluctuations is highest on climates characterized by a summer maximum in precipitation, like on Mt. Ortles, because of the proximity of air temperature to the melting point. Our calculated monthly estimates confirm the highest sensitivity of snow erosion during the warm and transition seasons, and low sensitivity during winter when air temperature is far from the melting point, even under the +3°C scenario (Fig. 12).

Based on the reported  $2 \pm 0.3$ °C warming experienced by the European Alps since 1850-1900 (Dumont et al., 2025, and references therein), the -2°C scenario should represent conditions at the end of the Little Ice Age (LIA) and the -3°C scenario could be representative of the coldest part of the LIA. Compared to these colder periods, current conditions lead to a calculated reduction of snow erosion ranging between 22% and 29% (Figs. 10 and 11). At the same time melt increased only from 0.2-0.3% to 2.9% of the total accumulated snowfall. Therefore, based on these calculations, the reduced snow erosion with increasing temperatures acts as negative feedback, increasing the annual accumulation rate at this site. Our calculations thus confirm the negative feedback deriving from a moderate temperature increase, which could increase the annual accumulation rate at high elevation in the Alps, as suggested for example by Haeberli and Alean, 1985.

A further small increase in air temperature compared to baseline conditions is expected to strongly overcome this negative feedback in the near future (IPCC, 2022). With just 1°C additional warming (already partially occurred since 2015 (Jaquemart et al., 2024)), there would be more than a fourfold increase in melt and only a 2% decrease in snow erosion. When combined with a 2.5% decrease in the fraction of solid precipitation, this would result in a significant decrease in the annual accumulation rate (Table 3, Fig.10). Therefore, a 'tipping point' for the wind erosion feedback is approaching at the AWS site. It is also worth noting that areas at lower elevation probably already experienced this critical temperature increase in the last decades and quickly transitioned from an erosion-dominated to a melt-dominated mass balance regime.

This specific study highlights the high sensitivity of glacier mass balance to snow erosion, and of snow erosion to air temperature fluctuations. These interactions may lead to a non-linear response of glacier mass balance to climate change, and

480

485

to negative feedbacks that should be taken into account when projecting the future behaviour of the alpine cryosphere at high elevation. However, similar interactions are likely to occur also in non-glacierized areas, affecting for example the thermal regime of the ground and the behaviour of the underlying permafrost.

We conclude that snow erosion processes and feedback should also be considered when paleoclimatic data recorded in the alpine cryosphere are used for reconstructing past environmental conditions such in the case of ice cores retrieved from high elevation glaciated areas. Similar to melt, wind erosion is potentially effective in removing a large fraction of the seasonal accumulated snow. On Mt. Ortles, this can occur especially during the coldest months, which are also the driest and windiest. In this case, for instance, a significant fraction of the isotopic winter signal might be missing from the Ortles ice core records. In addition, it is likely that during colder climatic phases, such as the LIA, an even lower winter 'imprint' is preserved in ice cores, because wind erosion increases at lower temperature.

For these reasons, we argue that proxy system models aimed at improving dating and interpretation of ice cores, could benefit from the inclusion of simple parameterizations of snow erosion, in addition to melt (e.g. Carturan et al., 2025). This refinement looks essential for a realistic modelling of the formation and preservation of paleoclimatic proxies such as the isotopic signal and its changes through time.

## **6 Conclusions**

In this study, we used the physics-based SNOWPACK model to investigate wind-driven snow erosion at a study site located at 3830 m a.s.l. on Mt. Ortles, in the Eastern Italian Alps, in close proximity to an ice core drilling site (Gabrielli, 2012). The model was run using high time resolution meteorological data collected by an automatic weather station that was operated between 2011 and 2015, combined with detailed mass balance observations carried out during the same period.

The results of this study highlight that snow erosion by wind is a principal factor affecting the mass balance at this highelevation site, because it removes 20.9% of the total snowfall when compared to 2.9% removal by melt. Therefore, under the current climatic conditions, wind-driven snow erosion represents the dominant ablation process at the study site of Mt. Ortles at 3830 m a.s.l.

Statistical analyses reveal an inverse relationship between air temperature and total snow erosion, because cold and dry snow is more susceptible to erosion when compared to wet snow. Overall, in the considered period, dry snow erosion accounts for 91% of the total erosion, whereas wet snow erosion accounts for the remaining 9%. Once the air temperature rises above the freezing point, wind-driven erosion becomes negligible.

For these reasons, snow erosion on Mt. Ortles is most effective during winter, when it averages 48% of the accumulated snow due to the interplay between high wind speed and low temperature. In the remaining part of the year, this amount ranges between 10% (in March and May) and 21% (in October).

Sensitivity analyses show that at 3830 a.s.l. on Mt. Ortles atmospheric warming leads to a significant decrease in the total snow erosion, which can be quantified in a 2% reduction in the amount of eroded snowfall per 1°C of warming. Sensitivity is

maximum in the warm season, reaching a 90% increase of snow erosion in August for a 3°C cooling, due to the higher proximity to the freezing point when compared to winter months.

Assuming a temperature increase of 2-3°C since the pre-industrial period, we calculated a 4.6-6.6% decrease in snow erosion (22-29% reduction in the eroded fraction of the total accumulated snowfall) and a 2.6-2.7% increase in melt. So far, this represented an important negative feedback at the study site, which, based on model results, experienced an increase in net accumulation. However, an additional 1°C atmospheric warming would be sufficient to overcome this negative feedback, because this site is approaching a transition from the erosion-dominated to a melt-dominated regime.

Although site-specific, these findings have broader implications for long-term glacier mass balance studies and, in particular, for paleoclimatic reconstructions from ice core data. Because wind erosion can remove a remarkable fraction of the seasonal snow signal, especially in winter, and because the efficiency of this process is likely to have varied over past decades/centuries in response to temperature fluctuations, we suggest considering this process in climatic and environmental reconstructions from ice core data. Explicitly accounting for snow erosion (in addition to melt), in the development of proxy system models will help to interpret the paleoclimatic signal preserved in alpine ice core archives.

#### APPENDIX A: Modelled monthly results for different air temperature scenarios

Table A1: Modelled monthly precipitation, snowfall, erosion, and melt (in mm and %) for the baseline scenario. Percentages are relative to total monthly precipitation.

| Month     | Precipitation (mm) | Snowfall (mm) | Snowfall (%) | Erosion (mm) | Erosion (%) | Melt (mm) | Melt (%) |
|-----------|--------------------|---------------|--------------|--------------|-------------|-----------|----------|
| January   | 399.02             | 391.64        | 98.15        | 151.66       | 38.72       | 0.00      | 0.00     |
| February  | 389.98             | 382.58        | 98.10        | 174.64       | 45.65       | 0.00      | 0.00     |
| March     | 303.92             | 298.18        | 98.11        | 31.73        | 10.64       | 0.00      | 0.00     |
| April     | 570.20             | 558.35        | 97.92        | 76.40        | 13.68       | 0.01      | 0.00     |
| May       | 327.50             | 321.80        | 98.26        | 30.25        | 9.40        | 2.79      | 0.87     |
| June      | 400.18             | 389.43        | 97.31        | 70.01        | 17.98       | 11.25     | 2.89     |
| July      | 793.62             | 768.16        | 96.79        | 99.97        | 13.01       | 18.37     | 2.39     |
| August    | 699.14             | 652.41        | 93.31        | 113.48       | 17.39       | 134.14    | 20.56    |
| September | 506.77             | 496.78        | 98.03        | 63.93        | 12.87       | 3.61      | 0.73     |
| October   | 711.48             | 697.60        | 98.05        | 150.13       | 21.52       | 3.54      | 0.51     |
| November  | 756.74             | 742.61        | 98.13        | 119.69       | 16.12       | 0.01      | 0.00     |
| December  | 283.43             | 278.23        | 98.16        | 166.50       | 59.84       | 0.00      | 0.00     |

Table A2: Modelled monthly precipitation, snowfall, erosion, and melt (in mm and %) for the -1 °C scenario. Percentages are relative to total monthly precipitation.

| Month     | Precipitation (mm) | Snowfall (mm) | Snowfall (%) | Erosion<br>(mm) | Erosion (%) | Melt (mm) | Melt (%) |
|-----------|--------------------|---------------|--------------|-----------------|-------------|-----------|----------|
| January   | 399.02             | 391.14        | 98.03        | 168.44          | 43.06       | 0.00      | 0.00     |
| February  | 389.98             | 382.57        | 98.10        | 199.06          | 52.03       | 0.00      | 0.00     |
| March     | 303.92             | 298.04        | 98.07        | 36.78           | 12.34       | 0.00      | 0.00     |
| April     | 570.20             | 558.59        | 97.96        | 79.93           | 14.31       | 0.00      | 0.00     |
| May       | 327.50             | 321.84        | 98.27        | 35.79           | 11.12       | 1.44      | 0.45     |
| June      | 400.18             | 392.30        | 98.03        | 72.82           | 18.56       | 6.52      | 1.66     |
| July      | 793.62             | 776.76        | 97.88        | 108.62          | 13.98       | 10.99     | 1.42     |
| August    | 699.14             | 669.30        | 95.73        | 142.32          | 21.26       | 15.03     | 2.25     |
| September | 506.77             | 496.78        | 98.03        | 72.87           | 14.67       | 1.96      | 0.40     |
| October   | 711.48             | 697.60        | 98.05        | 163.14          | 23.39       | 1.70      | 0.24     |
| November  | 756.74             | 742.66        | 98.14        | 142.52          | 19.19       | 0.00      | 0.00     |
| December  | 283.43             | 278.24        | 98.17        | 174.01          | 62.54       | 0.00      | 0.00     |

Table A3: Modelled monthly precipitation, snowfall, erosion, and melt (in mm and %) for the -2 °C scenario. Percentages are relative to total monthly precipitation.

| Month     | Precipitation (mm) | Snowfall (mm) | Snowfall (%) | Erosion<br>(mm) | Erosion (%) | Melt (mm) | Melt (%) |
|-----------|--------------------|---------------|--------------|-----------------|-------------|-----------|----------|
| January   | 399.02             | 391.24        | 98.05        | 181.46          | 46.38       | 0.00      | 0.00     |
| February  | 389.98             | 382.47        | 98.07        | 209.66          | 54.82       | 0.00      | 0.00     |
| March     | 303.92             | 298.21        | 98.12        | 38.87           | 13.03       | 0.00      | 0.00     |
| April     | 570.20             | 558.62        | 97.97        | 83.65           | 14.97       | 0.00      | 0.00     |
| May       | 327.50             | 321.72        | 98.24        | 41.09           | 12.77       | 0.88      | 0.27     |
| June      | 400.18             | 392.89        | 98.18        | 76.11           | 19.37       | 3.27      | 0.83     |
| July      | 793.62             | 778.69        | 98.12        | 119.22          | 15.31       | 3.69      | 0.47     |
| August    | 699.14             | 674.07        | 96.41        | 180.83          | 26.83       | 10.54     | 1.56     |
| September | 506.77             | 496.79        | 98.03        | 85.53           | 17.22       | 1.09      | 0.22     |
| October   | 711.48             | 696.47        | 97.89        | 176.95          | 25.41       | 0.74      | 0.11     |
| November  | 756.74             | 743.19        | 98.21        | 163.99          | 22.07       | 0.00      | 0.00     |
| December  | 283.43             | 278.57        | 98.29        | 178.02          | 63.91       | 0.00      | 0.00     |

Table A4: Modelled monthly precipitation, snowfall, erosion, and melt (in mm and %) for the -3 °C scenario. Percentages are relative to total monthly precipitation.

| Month     | Precipitation (mm) | Snowfall (mm) | Snowfall (%) | Erosion<br>(mm) | Erosion (%) | Melt (mm) | Melt (%) |
|-----------|--------------------|---------------|--------------|-----------------|-------------|-----------|----------|
| January   | 399.02             | 390.87        | 97.96        | 182.87          | 46.79       | 0.00      | 0.00     |
| February  | 389.98             | 382.92        | 98.19        | 205.93          | 53.78       | 0.00      | 0.00     |
| March     | 303.92             | 297.89        | 98.01        | 39.84           | 13.37       | 0.00      | 0.00     |
| April     | 570.20             | 558.45        | 97.94        | 84.86           | 15.20       | 0.00      | 0.00     |
| May       | 327.50             | 322.52        | 98.48        | 41.29           | 12.80       | 0.40      | 0.12     |
| June      | 400.18             | 392.81        | 98.16        | 78.42           | 19.96       | 2.02      | 0.51     |
| July      | 793.62             | 779.20        | 98.18        | 128.57          | 16.50       | 1.96      | 0.25     |
| August    | 699.14             | 683.58        | 97.77        | 227.48          | 33.28       | 8.81      | 1.29     |
| September | 506.77             | 496.70        | 98.01        | 92.53           | 18.63       | 0.52      | 0.10     |
| October   | 711.48             | 696.36        | 97.88        | 188.06          | 27.01       | 0.37      | 0.05     |
| November  | 756.74             | 743.32        | 98.23        | 174.37          | 23.46       | 0.00      | 0.00     |
| December  | 283.43             | 277.65        | 97.96        | 179.89          | 64.79       | 0.00      | 0.00     |

Table A5: Modelled monthly precipitation, snowfall, erosion, and melt (in mm and %) for the +1 °C scenario. Percentages are relative to total monthly precipitation.

| Month     | Precipitation (mm) | Snowfall (mm) | Snowfall (%) | Erosion<br>(mm) | Erosion (%) | Melt (mm) | Melt (%) |
|-----------|--------------------|---------------|--------------|-----------------|-------------|-----------|----------|
| January   | 399.02             | 391.52        | 98.12        | 139.60          | 35.66       | 0.00      | 0.00     |
| February  | 389.98             | 383.06        | 98.23        | 156.05          | 40.74       | 0.00      | 0.00     |
| March     | 303.92             | 298.17        | 98.11        | 25.50           | 8.55        | 0.00      | 0.00     |
| April     | 570.20             | 558.43        | 97.94        | 74.75           | 13.39       | 0.10      | 0.02     |
| May       | 327.50             | 320.94        | 98.00        | 25.32           | 7.89        | 3.33      | 1.04     |
| June      | 400.18             | 372.66        | 93.12        | 60.43           | 16.22       | 45.07     | 12.09    |
| July      | 793.62             | 704.72        | 88.80        | 81.76           | 11.60       | 204.46    | 29.01    |
| August    | 699.14             | 625.88        | 89.52        | 88.45           | 14.13       | 468.78    | 74.90    |
| September | 506.77             | 490.09        | 96.71        | 56.03           | 11.43       | 6.78      | 1.38     |
| October   | 711.48             | 697.28        | 98.00        | 137.27          | 19.69       | 5.37      | 0.77     |
| November  | 756.74             | 743.21        | 98.21        | 95.89           | 12.90       | 0.03      | 0.00     |
| December  | 283.43             | 277.78        | 98.01        | 158.07          | 56.90       | 0.00      | 0.00     |

Table A6: Modelled monthly precipitation, snowfall, erosion, and melt (in mm and %) for the +2 °C scenario. Percentages are relative to total monthly precipitation.

| Month     | Precipitation (mm) | Snowfall (mm) | Snowfall (%) | Erosion (mm) | Erosion (%) | Melt (mm) | Melt (%) |
|-----------|--------------------|---------------|--------------|--------------|-------------|-----------|----------|
| January   | 399.02             | 391.35        | 98.08        | 119.75       | 30.60       | 0.00      | 0.00     |
| February  | 389.98             | 382.62        | 98.11        | 122.57       | 32.03       | 0.00      | 0.00     |
| March     | 303.92             | 298.34        | 98.16        | 18.57        | 6.22        | 0.01      | 0.00     |
| April     | 570.20             | 558.14        | 97.89        | 60.88        | 10.91       | 0.42      | 0.08     |
| May       | 327.50             | 317.13        | 96.83        | 21.61        | 6.81        | 4.47      | 1.41     |
| June      | 400.18             | 339.03        | 84.72        | 54.70        | 16.13       | 226.89    | 66.92    |
| July      | 793.62             | 554.55        | 69.88        | 54.89        | 9.90        | 720.78    | 129.98   |
| August    | 699.14             | 455.89        | 65.21        | 56.74        | 12.45       | 800.71    | 175.64   |
| September | 506.77             | 482.03        | 95.12        | 51.29        | 10.64       | 23.83     | 4.94     |
| October   | 711.48             | 696.19        | 97.85        | 119.81       | 17.21       | 7.29      | 1.05     |
| November  | 756.74             | 743.97        | 98.31        | 80.70        | 10.85       | 0.20      | 0.03     |
| December  | 283.43             | 277.66        | 97.96        | 147.60       | 53.16       | 0.00      | 0.00     |

Table A7: Modelled monthly precipitation, snowfall, erosion, and melt (in mm and %) for the +3 °C scenario. Percentages are relative to total monthly precipitation.

| Month     | Precipitation (mm) | Snowfall (mm) | Snowfall (%) | Erosion (mm) | Erosion (%) | Melt (mm) | Melt (%) |
|-----------|--------------------|---------------|--------------|--------------|-------------|-----------|----------|
| January   | 399.02             | 391.50        | 98.12        | 108.29       | 27.66       | 0.00      | 0.00     |
| February  | 389.98             | 382.30        | 98.03        | 96.11        | 25.14       | 0.00      | 0.00     |
| March     | 303.92             | 298.07        | 98.07        | 12.46        | 4.18        | 0.13      | 0.04     |
| April     | 570.20             | 557.81        | 97.83        | 54.04        | 9.69        | 1.41      | 0.25     |
| May       | 327.50             | 307.78        | 93.98        | 19.55        | 6.35        | 4.66      | 1.51     |
| June      | 400.18             | 305.29        | 76.29        | 44.60        | 14.61       | 567.30    | 185.82   |
| July      | 793.62             | 459.47        | 57.90        | 30.70        | 6.68        | 1238.56   | 269.56   |
| August    | 699.14             | 286.19        | 40.93        | 34.01        | 11.88       | 1177.68   | 411.51   |
| September | 506.77             | 434.08        | 85.66        | 45.10        | 10.39       | 78.02     | 17.97    |
| October   | 711.48             | 690.84        | 97.10        | 107.09       | 15.50       | 15.61     | 2.26     |
| November  | 756.74             | 743.42        | 98.24        | 70.73        | 9.51        | 1.47      | 0.20     |
| December  | 283.43             | 278.60        | 98.29        | 143.01       | 51.33       | 0.00      | 0.00     |

540

## **APPENDIX B: Missing wind speed records**

Table B1: Monthly percentage of missing 15-minutes wind speed records. The total represents the percentage over the entire dataset.

| Month     | Missing wind speed records (%) |
|-----------|--------------------------------|
| January   | 12                             |
| February  | 0                              |
| March     | 0                              |
| April     | 0                              |
| May       | 22                             |
| June      | 22                             |
| July      | 2                              |
| August    | 5                              |
| September | 0                              |
| October   | 2                              |
| November  | 8                              |
| December  | 11                             |
| Total     | 7                              |

Data availability. The datasets from this study are publicly available at https://doi.org/10.5281/zenodo.15669722 (Carturan, 2025). The data files are stored in CSV format.

Author contribution. TLZ, LC, and ML designed the methodological approach. PG, LC, and GDF carried out the fieldwork. TLZ processed the meteorological data. TLZ ran the model with contributions from ML, NW, and MB. TLZ and LC performed the erosion analysis. TLZ and LC prepared the first draft. All authors contributed to the discussion and the final version of the manuscript.

Competing interests. The contact author has declared that none of the authors has any competing interests.

Acknowledgements. TLZ acknowledges the support and hospitality of the WSL Institute for Snow and Avalanche Research SLF (Davos, Switzerland) for hosting her during her PhD research stay. The authors are grateful to all the students, technicians, and scientists who contributed to the field activities in the period from 2009 to 2016; the alpine guides of the Alpinschule of Solda; the helicopter companies Airway, Air Service Center, and Star Work Sky; and the Hotel Franzenshöhe for logistical support. The authors acknowledge the editor and reviewers for their comments and suggestions.

*Financial support*. The research was funded by the Italian MIUR Project (PRIN 2010-11), "Response of morphoclimatic system dynamics to global changes and related geomorphological hazards" (the local and national coordinators are Giancarlo Dalla Fontana and Carlo Baroni), and was carried out within the RETURN Extended Partnership and received funding from

the European Union NextGenerationEU (National Recovery and Resilience Plan – NRRP, Mission 4, Component 2, Investment 1.3 – D.D. 1243 2/8/2022, PE0000005). The core samples were obtained as part of the Mt Ortles Ice Core Project funded by NSF awards 1060115 and 1461422 with the logistical support of Ripartizione Protezione antincendi e civile of the Autonomous Province of Bolzano in collaboration with the Ripartizione Opere idrauliche e Ripartizione Foreste of the Autonomous Province of Bolzano and the Stelvio National Park. This is Ortles project publication 14 (https://www.ortles.org).

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
