# Peer review of "Air temperature partitioning of snow accumulation, erosion and melt: a regime shift occurring on Mt. Ortles (Eastern Italian Alps)"

_EGUsphere, 2025_

## Referee Comment (RC4)

The study investigates snow accumulation and wind-driven erosion on a high-altitude Alpine glacier using nivo-meteorological and mass balance observations collected on Mt. Ortles (Eastern Alps) and a physics-based snow model. The simulation results are compared with mass balance measurements and indicate that wind erosion is the dominant ablation process. According to the simulations, wind erosion is most effective in cold, dry winter conditions and is strongly influenced by air temperature.

Indeed, snow accumulation and wind-driven erosion processes are difficult to measure and model on mountain glaciers, and the article addresses scientific questions within the scope of TC. The measurement dataset covers a long period (4 years of 15-minute averages) and appears to be of high quality for a such challenging environment. However, the information provided on the measurements is incomplete (see comments below).

However, some questions arise regarding the methodology and arguments used to support the conclusions. I think the interpretation of the results is not really convincing, so that the discussion does not reach substantial conclusions. These points are detailed below. Thus, the manuscript does not provide much innovative inputs on the subject and does not represent substantial progress beyond current scientific understanding.

**Major comments**

**- Lack of validation of the snow model and blowing snow simulations**

In section 3.3, the text mentions that two "multiplicative factors for precipitation and wind speed were adjusted iteratively to minimize the Root Mean Square Error (RMSE) between measured and modelled mass balance at the simulation site" (lines 199-201). However, calibrating the model on a variable such as the mass balance, which results from all the (complex and interdependent) processes simulated in the snow, undermines the benefits of using a physical model due to error compensation issues. This physics-based snow model is based on equations of mass and energy conservation and the parameters should have a physical interpretation, so they can be linked to measurable physical quantities. Such a calibration procedure on the mass balance breaks the links between the parameters and the physical processes or quantities they are supposed to represent.

A multiplicative factor for precipitation measured in the valley at Solda of 1.55 (Section 4.1) makes sense as we can expect higher precipitation amounts in the mountains (however a discussion about the value of the parameter is missing). Precipitation is an important input data in snowpack mass balance simulations, but little is known about it in high mountain areas. Are there other measurements that could be used to better constrain this precipitation correction factor? For example, mass balance measurements in snow pits dug in areas of the glacier less exposed to snow erosion?

The use of a calibrated multiplicative factor for wind speed is even more questionable. The authors argue: "... the wind speed was also adjusted by means of a multiplicative factor, to account for the fact that on top of Mt. Ortles the wind is not expected to follow strictly a vertical logarithmic profile". As, if I correctly understand, this multiplicative factor is applied to the wind speed measured by the weather station on the glacier, I don't understand why this measurement needs such a correction. If the authors are referring to stability corrections of neutral logarithmic wind profiles, this correction is certainly small for measurement heights below 2 m (the measurement height is an important information that is missing in the

manuscript). In addition, the study focuses on strong wind conditions associated with blowing snow events, during which wind profiles are close to neutral. As a result of the calibration, the multiplicative factor for wind speed is fixed to 0.70 (Section 4.1). the authors should use physical arguments to explain this 30% reduction in measured wind speed (or one might think that this factor is used to compensate for other simulation errors).

The use of these two correction factors should be examined in more detail and, for example, the sensitivity of the model results to these parameters could be investigated.

The validation of the snow model results relies on Table 2 comparing observed and modelled snow water equivalent, but this table is difficult to read a figure would be easier to understand. Furthermore, the titles of Table 2 are not clear to me. I do not understand the cumulated balance: shouldn't it be the cumulated sum of the data in the previous column? And given the expected measurement uncertainties, is it appropriate to display the mass balance values to the nearest millimeter w.e.?

Given that the conclusions of the article are based primarily on the results of the model, the inappropriate method of validation of the snow model is a critical problem in this study.

- *Lack of information on the meteorological measurements and calculation methods of the surface energy fluxes.*

In Section 3.1., very little information is given concerning the measurements of the meteorological variables. The text mentions that details on sensors and measured data can be found in Carturan et al. (2023) but this data paper only details the air temperature measurements. Thus, information is missing about the sensor characteristics and accuracy, measurement height and so on… The methods of data processing, corrections and gap-filling should also be presented. This information on the snow model input data is necessary for a correct interpretation of the model results (particularly in terms of uncertainties).

I suppose that the sensor measurements (in particular, wind speed, air temperature and humidity) are corrected according to the changes in height above the snow surface (measured by the SR50). This is an important correction that should be detailed.
In fact, the authors do not use the SR50 measurements. Indeed, interpreting these measurements is difficult in very windy environments. However, as very little information is available on snowfall and snow erosion processes, some useful information could certainly be derived from these measurements, for example on the timing of snowfall or snow erosion events.

To get an idea of the meteorological context, it would also be useful to see time series of some meteorological variables (e.g., temperature, wind speed, radiation fluxes).

The calculation methods of the surface energy fluxes must be presented, particularly for the turbulent fluxes. I suppose that a bulk-aerodynamic profile method is applied. In this case, how are determined the roughness lengths for momentum, temperature and humidity? Are these parameters important in the blowing snow simulations? This information is also necessary to interpret the sensitivity analysis of snow erosion to changes in air temperature (Section 3.5).

- *Insufficient interpretation of results and too vague discussion*

The interpretation of the modelling results is sometimes insufficient. For instances:
- Due to the large spread of data, Figure 5 is difficult to interpret.
For instance, for the no snowfall cases, does it make sense that the fitted curve for wet snow (orange) is above that for dry snow (light blue)? At the same wind speed, would the "w.e. eroded" be larger for wet snow than for dry snow? Furthermore, the numbers of dry/wet conditions during/after snowfall are certainly very different (these numbers should be indicated), which complicates the statistical interpretation of the graph. I think that this graph provides no useful information.
- Lines 289-290 and Figure 6: For wet erosion events, what could explain such different cumulative relative frequency distributions of snow erosion events and cumulative relative SWE eroded? Can the sudden increase in "swe eroded" when the age reaches 900-1000 hours be explained?
- p.17, Figure 7: for dry snow, why the frequency of SWE eroded peaks at larger wind speed that the frequency peak of snow erosion? what exactly represents "snow erosion" in Figure 7? Units?
- There are very few wet snow erosion events: "dry snow erosion accounts for 91% of the total modelled erosion" (lines 292-293). Are the statistical results for wet snow events presented in Figures 7, 8, and 9 significant? In particular, how can the two peaks in the frequency distribution of the threshold wind speed for wet snow erosion shown in Figure 9 be explained?

The discussion remains rather vague (especially in Section 5.1), mainly due to the lack of field observations concerning snow accumulation and erosion processes (see comments on snow model validation issues).

The text states (p.28-29): "This specific study highlights the high sensitivity of glacier mass balance to snow erosion, and of snow erosion to air temperature fluctuations. These interactions may lead to a non-linear response of glacier mass balance to climate change and to negative feedbacks that should be taken into account when projecting the future behaviour of the alpine cryosphere at high elevation".
However, this point-scale study indicates local snow loss. Eroded snow can be transported and deposited elsewhere on the glacier, so information is missing to consider the local eroded snow as a loss in the glacier mass balance.

*- The analysis of the sensitivity analysis of snow accumulation and erosion to changes in air temperature is not convincing and does not provide much innovative insights.*

There is insufficient information on how changes in air temperature are taken into account in the simulations. For example, does air humidity remain unchanged in relative or specific values? How are calculated the turbulent fluxes (see comment above)? It is important to know whether a stability correction is applied in the calculations of the turbulent fluxes (negative feedback).

Since the scenarios presented here only take into account changes in air temperature (all other meteorological variables remain unchanged), the physical consistency between the input meteorological data is lost and the benefit of using a physics-based model is greatly reduced. The most problematic point is certainly not taking into account changes in precipitation for such significant changes in air temperature (up to 3°C). For instance, the authors consider that "the -2°C scenario should represent conditions at the end of the Little Ice Age (LIA) and the -3°C scenario could be representative of the coldest part of the LIA" (p.28). However, there

are numerous studies on glacier extents during the LIA in the Alps, and they generally indicate an increase in precipitation (e.g., Vincent et al., 2005). Thus, it is probably not fair to consider that the -2°C or -3°C scenario should represent conditions of the LIA.

**Other comments**

- Section 3.4: please clarify the definition of "the life" of a snow layer.
If I understand correctly, the "life" of a snow layer ends when it is removed by wind erosion and it starts when "snow accumulation exceeded 0.1 mm h-1" (Lines 210-211). Thus, the life of a snow layer should also end due to a snow accumulation event (not only due to snow erosion). However, the interpretation of the results (e.g. Section 4.2.1) seems to consider that the variable "Age" is only limited by snow erosion (?). This point should be clarified. Furthermore, the term "life" is not appropriate to characterize a snow layer (it is not a living being); a term such as "duration" would be more appropriate.

The wording is not always specific enough, for instances:
- too much use of the names of the simulation variables in the text (e.g. MS_wind, WS_Th…), it's not easy to understand, especially in the results section
- Figure 1 should be cited in the first paragraph of the Section 2 (Study Area)
- line 340: Figure 12 should be cited before, at the beginning of the paragraph.

**Specific comments:**

- Line 53: "this work": self-citation?
- Line 104-105: also due to the influence of spatial variations in precipitation.
- Lines 183-184: I do not understand what the fetch length represents here.

**Tables and Figures**

Figure 10: the lowest panel on the left showing the baseline scenario is not useful (it is shown in all the other panels).

I'm not sure that Table 3 is necessary. Furthermore, is it appropriate to show precipitation, snowfall, or melt amounts in mm (w.e.?) with two decimal digits?

**References**

Vincent, C., Le Meur, E. & Six, D. Solving the paradox of the end of the Little Ice Age in the Alps. Geophysical Research Letters 32, doi:10.1029/2005GL022552 (2005).